# A Comparison between Enrichment Optimization Algorithm (EOA)-Based and Docking-Based Virtual Screening

**DOI:** 10.3390/ijms23010043

**Published:** 2021-12-21

**Authors:** Jacob Spiegel, Hanoch Senderowitz

**Affiliations:** Department of Chemistry, Bar-Ilan University, Ramat-Gan 5290002, Israel; spiegel.jacob@gmail.com

**Keywords:** enrichment optimization algorithm, docking, virtual screening, QSAR, Glide, GOLD, AutoDock Vina

## Abstract

Virtual screening (VS) is a well-established method in the initial stages of many drug and material design projects. VS is typically performed using structure-based approaches such as molecular docking, or various ligand-based approaches. Most docking tools were designed to be as global as possible, and consequently only require knowledge on the 3D structure of the biotarget. In contrast, many ligand-based approaches (e.g., 3D-QSAR and pharmacophore) require prior development of project-specific predictive models. Depending on the type of model (e.g., classification or regression), predictive ability is typically evaluated using metrics of performance on either the training set ($e.g.,Q_{\mathrm{CV}}^{2}$) or the test set (e.g., specificity, selectivity or $Q_{F1/F2/F3}^{2}$). However, none of these metrics were developed with VS in mind, and consequently, their ability to reliably assess the performances of a model in the context of VS is at best limited. With this in mind we have recently reported the development of the enrichment optimization algorithm (EOA). EOA derives QSAR models in the form of multiple linear regression (MLR) equations for VS by optimizing an enrichment-based metric in the space of the descriptors. Here we present an improved version of the algorithm which better handles active compounds and which also takes into account information on inactive (either known inactive or decoy) compounds. We compared the improved EOA in small-scale VS experiments with three common docking tools, namely, Glide-SP, GOLD and AutoDock Vina, employing five molecular targets (acetylcholinesterase, human immunodeficiency virus type 1 protease, MAP kinase p38 alpha, urokinase-type plasminogen activator, and trypsin I). We found that EOA consistently outperformed all docking tools in terms of the area under the ROC curve (AUC) and EF_1%_ metrics that measured the overall and initial success of the VS process, respectively. This was the case when the docking metrics were calculated based on a consensus approach and when they were calculated based on two different sets of single crystal structures. Finally, we propose that EOA could be combined with molecular docking to derive target-specific scoring functions.

## 1. Introduction

Time and money are two of the most required resources in the design of new drugs and materials. Several techniques are available to expedite and lower the costs of these processes, such as functional genomics [1], high-throughput screening (HTS) [2] and combinatorial chemistry [3]. Over the years, computational methods have demonstrated their ability to complement and even replace experimental techniques for such tasks.

Among the computational methods, virtual screening (VS) stands out as a viable alternative to HTS in the initial stages of drug and material design projects. VS is typically performed using structure-based approaches employing molecular docking, and to a lesser extent, simulation methods, or various ligand-based approaches [4,5,6]. Most docking tools, scoring functions and force fields utilized in structure-based VS were designed to be as general as possible, as evident, e.g., from the large volume of experimental data used for their derivation [7,8,9,10,11], and consequently only require knowledge of the 3D structure of the biotarget, although additional experimental information, when available, could be used to select the most appropriate tool(s) and to optimize their performances. In contrast, many ligand-based approaches require the prior development of target-specific predictive models by using information on active (and when available inactive) compounds. Ligand-based approaches for VS are typically implemented by means of pharmacophore models [5], 3D-QSAR [12,13] or various similarity search strategies [14,15]. Both structure-based and ligand-based approaches have demonstrated notable success in multiple VS campaigns [16,17].

A particularly appealing yet somewhat less common approach to virtual screening is presented by QSAR equations derived from easy to calculate 1D, 2D and sometimes global 3D descriptors. Several such studies were reported in the literature, and in most cases, the descriptors were calculated for the ligands in their unbound states [18,19,20,21,22,23,24,25,26,27,28]. Some of these efforts were summarized in several review articles [29,30]. In other cases, ligand-based descriptors were combined with descriptors derived from the ligands’ 3D conformations as obtained from molecular docking [31,32] or with the docking scores themselves [33]. However, these models were typically not used for virtual screening due to the computational resources required for large scale docking. An interesting combination of QSAR and docking for the purpose of VS was recently presented by Gentile et al., who used a deep neural network based on molecular fingerprints to predict docking scores of >1.3 billion compounds retrieved from the ZINC database [34]. This method was subsequently used for the VS of a similar number of compounds against the SARS-CoV-2 main protease [35]. All in all, QSAR-based VS holds promise for handling the increasingly large collections of commercially available or synthetically feasible compound collections offered by many vendors. This is because the computational resources needed for calculating such descriptors are significantly less than those required by other techniques.

A common theme of all the above-mentioned literature reports is that QSAR models derived for VS were validated using metrics of performance on either a training set ($e.g.,Q_{\mathrm{CV}}^{2}$) or a test set (e.g., metrics derived from the confusion matrix for classification models or $Q_{F1/F2/F3}^{2}$ for regression models) [36]. In this respect, it is important to note the lack of correlation between internal and external validation, a phenomenon sometimes referred to as the “Kubinyi Paradox” [37,38]. This, together with similar observations, have led to the realization that models should be evaluated on external test sets only [39]. However, irrespective of the exact nature of the evaluation metric, there is no reason to a priori assume that any of these metrics could reliably assess the performances of a QSAR model in the context of VS. This is because the task faced by VS, namely, the identification of a set of weakly active compounds from within a large pool of diverse, mostly inactive compounds, is quite different from the ability to qualitatively or quantitatively predict the activities of a small set of similar compounds. Thus, we argue that the evaluation of QSAR equations (and by extension of any computational model) should reflect their intended usage. In particular, if QSAR equations are derived with VS in mind, they should be evaluated in a VS scenario. Moreover, if the derivation of QSAR equations is treated as an optimization problem in the space of the molecular descriptors, which is often the case, then for the purpose of VS, the metric to be optimized should be VS-aware.

With this in mind, we previously presented the enrichment optimization algorithm (EOA), which derives QSAR models in the form of multiple linear regression (MLR) equations by optimizing an enrichment-like metric, and demonstrated its superiority in small-scale VS campaigns over QSAR equations derived by optimizing according to a “classical” metric (mean averaged error) [40]. Still, the original EOA algorithm suffered from several drawbacks, and in particular from a high degree of redundancy in the optimized metric. Thus, many QSAR equations led to identical values of the evaluation metric with no way to further rank them.

In this work, we present an improved version of EOA and demonstrate its superior performances in the virtual screening of five protein targets, this time in comparison with the most common VS approach, namely, molecular docking.

## 2. Results

Table 1 presents the results obtained with the EOA models for all datasets, whereas Table 2 presents a comparison between the results obtained with EOA and all the docking tools tested in this work for all test sets. Test set performances were evaluated using the AUC and EF_1%_ metrics. The docking results are based on the consensus approach across two crystal structures for each target. The results for the individual crystal structures are presented in Appendix A.

The EOA results presented in Table 1 demonstrate the expected yet small decrease in performance when going from the training sets to the validation sets (overall averaged percentages of active compounds found within the first *L* places of the ranked list of 85% and 82%, for the training sets and validation sets, respectively). A much larger decrease occurred for the test sets (average percentage = 41%), which could be attributed to the much smaller percentage of active compounds in these sets (see Materials and Methods section). In terms of the number of descriptors, we note a slight increase in performance when going from 7-descriptor to 10-descriptor and to 13-descriptor models. However, this increase was consistent across training, validation and test sets: The averaged percentages of active compounds found within the first *L* places of the ranked list were 83%, 86%, and 87% for training sets calculated with 7, 10 and 13-descriptor models. The corresponding scores for the validation sets were 81%, 83% and 84%; and for test sets, 40%, 42% and 42%. Taken together, these results suggest that our models are unlikely to be over-fitted. We note that in all cases, the number of descriptors in the final models was well below the number of compounds used for model derivation (see Table 1). Finally, in terms of the different protein targets, the best results were obtained for UROK and TRY1, followed by HIVPR, whereas models derived for ACES and MK14 gave overall poorer results (averaged percentages of active compounds found within the first *L* places of the ranked list obtained for training, validation and test sets were: UROK: 96%, 94%, 58%; TRY1: 91%, 88%, 62%; HIVPR: 87%, 86%, 34%; ACES: 76%, 69%, 28%; MK14: 76%, 74%, 26%). A list of the most common descriptors appearing in all EOA equations derived for each dataset together with the number of occurrences and a short explanation is provided in Appendix A. A complete list of descriptors is provided in Appendix A. 

The results of the VS are presented in Table 2. Similarly to Table 1, we see a slight increase in EOA performance when going from 7-descriptor to 10-descriptor models but no further increase when going to 13-descriptor models (averaged AUC and EF_1%_ values across all EOA models across all datasets and subsets for 7, 10 and 13-descriptor models were 0.91 and 42.4; 0.94 and 48.8; and 0.95 and 49.0, respectively). In terms of the different protein targets, the best average AUC and EF_1%_ values were obtained for UROK and TRY1, followed by the other protein targets (UROK: 0.98 and 76.5; TRY1: 0.98 and 75.3; MK14: 0.91 and 30.2; ACES: 0.89 and 28.4; HIVPR: 0.91 and 23.2). This trend is similar although not identical to that observed in Table 1.

Most significantly, the results in Table 2 clearly indicate that the EOA algorithm outperformed all docking programs tested in this study across all subsets and datasets in terms of both AUC and EF_1%_ values, even when the latter were based on the consensus approach (averaged AUC across all EOA models across all datasets and subsets: 0.93; averaged AUC across all docking tools across all datasets and subsets: 0.76 for the consensus approach, 0.73 based on the DUD-E associated structures (Appendix A) and 0.74 based on the alternative structures (Appendix A); the corresponding EF_1%_ values are 46.7, 17.8, 15.3, and 16.2 for EOA, consensus docking, DUD-E associated structures and alternative structures, respectively). Thus, while using a consensus approach across two structures improved the docking results relative to using a single structure, the performances of EOA were still better. 

Finally, while a comparison between the different docking algorithms was not the focus of this study, we note that in terms of AUC values and using the consensus approach, AD Vina performed the best for ACES, HIVPR and MK14. For UROK and TRY1, the results are test set-dependent, with Glide and Gold outperforming AD Vina. In terms of EF_1%_, GOLD performed the best for ACES and HIVPR, whereas Glide performed the best for MK14, UROK and TRY1. Thus, no clear-cut trends were observed. A comparison of Table 2 with Appendix A suggests that out of the five targets considered in this work, the largest increases in AUC and EF_1%_ upon moving from single crystal-based VS to consensus-based VS occurred for the TRY1 dataset docked with Glide, and for the MK14 dataset also docked with Glide, respectively. 

Figure 1 presents, for each dataset, the EOA and docking-generated ROC curves, based on the consensus approach. As noted in the Materials and Methods section, each dataset gave rise to four subsets which led to four corresponding test sets. Thus, for each dataset, we chose as a reference the test set that yielded the best ROC curve for any of the docking programs and compared the results obtained with the other docking tools and with the EOA algorithm to it. A similar analysis based on using as a reference the test set that yielded the best ROC curve across the two crystal structures (DUD-E-associated and alternative structures) and the four test sets is presented in Appendix A. The complete set of ROC curves for both sets of structures and for the consensus approach is provided in Appendix A. These results reinforce those presented in Table 2, Appendix A—namely, that for this set of targets, the EOA algorithm performed the best. Note that this is the case even though the EOA algorithm is not necessarily represented by its best result. For example, for the UROK case we chose to present the results obtained for set 3, since for this set, we obtained the best results from among the docking tools with GOLD (AUC = 0.86). The results obtained with the EOA algorithm for this set were AUC = 0.96. However, the results obtained with EOA for set 1 of this target were slightly superior, at 0.997.

To further check that the results obtained with the EOA algorithm were not chance-correlated, we performed Y-scrambling. For this purpose, the training set that gave rise to the best EOA model from among the four subsets for each of the five parent datasets was scrambled five times. Each scrambling was performed by tagging all active compounds as inactive and by randomly tagging the appropriate number of inactive compounds as active. Next, EOA models were derived from the scrambled datasets using 10 descriptors for the ACES, HIVPR and TRY1 datasets, and 13 descriptors for the MK14 and UROK datasets, and were applied to the test sets. The resulting ROC curves are presented in Figure 2 and demonstrate performances well below random selection. Thus, the original models derived from the unscrambled dataset are likely not chance-correlated.

## 3. Discussion

In this work we present an improved version of our recently reported enrichment optimization algorithm (EOA). EOA derives QSAR models by optimizing an enrichment-like function, specifically by ranking a set of *L* active and *O* inactive compounds using an MLR equation and by maximizing the number of active compounds within the top *L* places of the ranked list. In this version we have augmented the scoring function by a secondary score which favors solutions in which active compounds, if found beyond the first *L* places, and inactive compounds, if found within the first *L* places, occupy positions as close as possible to position *L*. However, we deliberately restricted this secondary score to take values in the range of (0, 1). This effectively means that solutions that introduce more active compounds into the first *L* places of the list will always be preferred irrespective of the positions of the other active/inactive compounds. We argue that this is a viable strategy for virtual screening, wherein the purpose is to maximize the number of active compounds at the top of a list ranked by some scoring function, although other strategies could also be considered. From within solutions with equal numbers of active compounds in *L*, the secondary score favors those that “push” inactive compounds towards the bottom of the *L* (active) list and active compounds towards the top of the *O* (inactive) list.

Using the new algorithm, EOA models were derived according to common best practices [41,42,43]. Specifically, models were derived using a training set and validated on independent validation and test sets. All divisions into training/validation/test sets were performed at random and repeated four times. The only difference between the validation and test sets were the proportions between active and decoy compounds, tests sets being constructed with a small percentage of active compounds (i.e., 0.5–0.7%) which is the common case for datasets used in virtual screening [44,45]. To make sure that our models are not over-fitted we derived 7, 10 and 13-descriptor models and compared their performances. To make sure our models are not chance-correlated, we performed y-scrambling. 

The results section suggests that the EOA models derived in this work are able to retrieve on average > 80% of the active compounds from training and validation sets and over 40% of the active compounds from test sets designed to include only a small fraction of active compounds, typical of VS campaigns. Importantly, these EOA models are unlikely to be chance-correlated or over-fitted. Looking at the most common descriptors appearing in the EOA equations derived for the different datasets (Appendix A), we note that many of them are directly relevant to the ligand-protein recognition process (e.g., number of N-H groups, number of hydrogen donor groups, electrotopological state indices, molecular charge descriptors) whereas others are more related to ligands’ ADME properties (e.g., ALOGP) or overall structure (e.g., Balaban index, Sum of topological distances between all nitrogen atoms in the molecule). All in all, the presence of these specific descriptors in the final EOA equations makes physical sense.

Having identified the most common descriptors, we wanted to see whether better EOA models could be developed using a subset of them and in particular such that emphasize protein-ligand interactions and key ADME ligand properties. For this purpose, we focused on a subset of seven descriptors, namely, PEOE3, HBD, MR1, ALOGP2, ALOGP7, ssCH2_Cnt and aaCH_Cnt (see Appendix A for a short explanation on their meaning) and constructed five EOA models, one for each target, using a total of three descriptors selected from this pool. The results are presented in Table 3 and are overall poorer than those obtained with the 7, 10 and 13-descriptor models (except of the 7-descriptors model for set 1 of the HIVPR target).

Next, we wanted to test whether the performances of EOA-based models are correlated in any way with the ease of differentiating between active and decoy compounds in the different datasets. As a surrogate to ease of separation, we chose to look at average distances between actives and decoys. Since actives/decoys distances depend on the specific compounds and on the descriptors used to characterize them, different distances are expected for each of the three EOA models derived for each of the four subsets from each target. Thus, for each target we calculated distances based on the descriptors selected for the best 10-descriptors EOA model. This is because for three out of the five targets (MK14, UROK, TRY1), EOA models based on ten descriptors performed the best, in terms of the AUC metric, across all four subsets. Furthermore, since EOA models were derived using normalized descriptors, for the purpose of relevant comparison, distance calculations also employed the normalized descriptors. For ease of calculation, Euclidian distances were calculated using all principle components derived from principle component analysis (PCA) performed with the WEKA program [46]. The results of our calculations together with the corresponding AUC values are presented in Table 4 and suggest that: (1) The average distances calculated for the different subsets are not significantly different from one another and (2) Overall, there is no correlation between the actives/decoys distances and model performances. Thus, it seems that the better performances observed, e.g., for the UROK and TRY1 sets are not necessarily the result of a larger separation between active and decoy compounds.

Finally, when compared against three common docking-based virtual screening tools, namely, Glide-SP, GOLD and AutoDock Vina EOA performances where significantly better both in terms of the AUC which informs on the overall success of the process and in terms of EF_1%_ values which inform on the success of the process in its initial stages. This was the case both when docking metrics were calculated based on the consensus approach or based on two different single crystal structures. 

The comparison between EOA and docking requires some discussion. Cleves et al., have advocated the usage of multiple crystal structures for virtual screening, demonstrating a modest increase in performances relative to the usage of a single crystal structure (averaged AUC increased from 0.81 ± 0.11 to 0.84 ± 0.09 for 92 targets [47]). Admittedly, some of the targets in the DUD-E^+^ database (including two utilized in this study, ACES and MK14) greatly benefited from the inclusion of multiple protein structures. In the case of ACES, we observed no increase in AUC on going from single structure-based VS to the consensus and only a small increase in EF_1%_ (average AUC across the three docking programs, four datasets and two single structure-based VS is 0.73 ± 0.05; the average AUC across the three docking programs and the four data sets for the consensus approach is 0.73 ± 0.06. The corresponding numbers for EF1% are 16.2 ± 8.2 and 17.6 ± 8.7). This could be attributed to the smaller number of structures used in our consensus (two vs. five). We note however, that the average AUC and EF_1%_ values obtained in this study for single structure-based VS, 0.73 and 16.2, are significantly higher than the numbers obtained by Cleves et al. for single structure-based VS (0.53 and 3) and in fact closely match the results of the five-structures consensus (0.74 and 19). Thus our single structure-based results leave much less room for improvement. The average (across all models and datasets) AUC and EF_1%_ values obtained by EOA are significantly higher at 0.89 ± 0.03 and 28 ± 19. In the case of MK14, we observed small increases in both AUC and EF_1%_ on going from single structure-based VS to the consensus (AUC: 0.68 ± 0.07 and 0.72 ± 0.04, respectively; EF_1%_: 8.9 ± 2.3 and 12.4 ± 5.0, respectively). For this target, the lowest AUC value was obtained with Glide using the alternative crystal structure (0.55 ± 0.01). This number significantly increased to 0.74 ± 0.01 upon going to the two structure-based consensus. Cleves et al. reported a similar increase from 0.66 to 0.89 using a five-structure consensus. Still, our EOA results are significantly higher at 0.92 ± 0.02 and 30.2 ± 8.1 for the AUC and EF_1%_, respectively. Finally, we note that while a consensus approach may benefit docking-based VS, in many projects, single structure-based VS is still the method of choice, either because of limited computational resources or because of lack of multiple crystal structures for the target of interest [34,35,48,49,50,51,52,53,54,55]. 

A possible explanation for the improved performances of EAO over docking is that none of the docking tools we used were specifically trained on any of the specific targets, whereas the EOA models were. Thus, the better performances obtained with EOA could be regarded as another manifestation of the superiority of local over global models. While we cannot forego this argument, we note that in our previous work we demonstrated that, in the context of VS, EOA-derived models outperformed other regression and classification models trained on the same set of data [40]. Thus, using an enrichment-aware function for model derivation clearly has merit for the purpose of virtual screening. 

Despite its good performances, two limitations of EOA and in fact of all ligand-based approaches for virtual screening should be noted: (1) Such approaches require information on active (and preferably also on inactive) compounds in order to derive and validate the models. While pharmacophore models could be developed using a rather small dataset, QSAR-based models typically require larger sets. EOA is no exception; however, since the method can utilize qualitative activity data, it has an advantage in this respect compared to regression-based models. In contrast, molecular docking could be performed with no a priori knowledge of active or inactive compounds. However, to validate and/or fine-tune the docking procedure for a specific target, this information is mandatory. In this respect, the difference between docking-based and EOA-based virtual screening is how to use available information, namely, for method validation and fine-tuning only (docking) or for method development and validation (EOA). (2) Ligand-based methods do not provide information on binding modes. EOA per se cannot overcome this limitation. However, in cases where information on active and inactive compounds is available, combining EOA with docking presents an appealing strategy whereby the components that make up the scoring function could be used as descriptors for the derivation of EOA-based models. This amounts to re-adjusting the weights of these components in a target-specific, virtual screening-aware manner. Depending on the specific docking tool, the new weights could be used either for pose re-scoring or even for the docking itself. In addition to providing target-specific scoring functions, this approach will also serve to more reliably compare between the performances of EOA and docking and to unveil the true advantage in using an enrichment-aware metric for model derivation by segregating the effect of the optimized function from that of model locality/generality. Work along these lines is currently being conducted in our laboratory. 

Despite these limitations, we view EOA-based models as viable and practical tools for virtual screening. While model derivation might be time consuming, particularly for large datasets characterized by multiple descriptors, using such models for virtual screening is highly efficient in terms of computational resources. Thus, such models can easily handle the currently available large collections of commercial and proprietary screening compounds, thereby increasing the probability of identifying good starting points for drug and material design efforts. 

## 4. Materials and Methods

### 4.1. Datasets

Datasets for five protein targets, namely, acetylcholinesterase (ACES), human immunodeficiency virus type 1 protease (HIVPR), MAP kinase p38 alpha (MK14), urokinase-type plasminogen activator (UROK) and trypsin I (TRY1), were retrieved from the DUD-E database [56]. These targets represent four different protein families according to the Pfam classifier [57] (carboxylesterase, retroviral aspartyl protease and P-kinase for ACES, HIVPR and MK14, respectively; and trypsin for UROK and TRY1). All datasets contain compounds which were experimentally determined to be active or decoy compounds. The PDB structures associated with each target in the DUD-E database are: 1e66, 1xl2, 2qd9, 1sqt and 2ayw, respectively. In addition, for each target we selected an alternative structure as suggested by the DUD-E^+^ database [47], namely, 1acj, 2pwc, 3o8t, 4fue and 3rxl. PDB codes, and the numbers of active and decoy compounds for each target are listed in Table 5. 

All compounds and proteins considered in this work were prepared by Schrodinger’s LigPrep program [58] and the Protein Preparation Wizard program [59], respectively. Protein preparation consisted of addition of hydrogen atoms, completion of missing side chains/residues and assignment of correct protonation states for ionizable residues. Ligand preparation consisted of obtaining reliable conformations, tautomeric forms and protonation states (at pH = 7).

Next, 1-dimensional and 2-dimentional (1D and 2D) molecular descriptors for all compounds from all datasets were calculated by the Canvas program [60,61]. The resulting ~750 descriptors (see Appendix A for a listing of the main descriptor types) were preprocessed by removing correlated ($r^{2}\;>\;0.7$), constant and nearly constant (i.e., constant over 70% of the compounds) descriptors. The remaining descriptors were normalized using z-score scaling. 

For the purpose of developing enrichment optimizer algorithm (EOA) models (see below), four subsets, each consisting of active and decoy compounds, were randomly selected from each parent dataset. These subsets were randomly divided into training and validation sets in an 80%/20% ratio, and the unselected compounds served as test sets. As a result of this selection procedure, the test sets contained much smaller percentages of active compounds (0.5–0.7%). This was done on purpose in order to mimic real VS campaigns. The compositions of all datasets are listed in Table 5.

### 4.2. Enrichment Optimizer Algorithm (EOA) Algorithm

In our previous work, we presented a novel algorithm for the derivation of multiple linear regression (MLR) equations suitable for usage in virtual screening, based on the optimization of an enrichment-like objective function. We termed the new algorithm enrichment optimizer algorithm (EOA) [40]. Briefly, EOA accepts as input a set of *L* active compounds together with a set of *O* inactive (either known inactive or decoy) compounds characterized by a set of *N* molecular descriptors. The algorithm then derives an MLR equation to rank the compounds, counts the number of active compounds within the first *L* places of the ranked list and then uses a Monte Carlo/simulated annealing (MC/SA) optimizer to maximize this number in the space of the descriptors and their weights. 

One drawback of the original EOA algorithm was the very limited range of values the objective function could take. This led to a high degree of redundancy manifested as multiple MLR equations leading to the same final value of the objective function with no means to further rank them. To address this problem, we introduced a post processing mechanism whereby redundant solutions with the highest value for the objective function were further scored based on the positions of active compounds that were ranked beyond the first *L* places and of inactive compounds that were ranked within the first *L* places. Higher (i.e., better) scores were allocated to solutions where both types of compounds had ranks closer to *L*. 

In the present study we generated a new scoring function by combining the number of active compounds located within the first *L* places of the ranked list (primary score) with the results of the above-described post processing mechanism (secondary score). The secondary score was normalized to be within the range of 0–1 by means of an inverse sigmoid function. This was done in order to give preference to solutions (i.e., MLR models) that gave rise to the maximal number of active compounds within the first *L* places, irrespective of the precise locations of other active or inactive compounds. As before, the new scoring function was optimized using MC/SA in the space of the descriptors and their weights. Figure 3 presents the flowchart of the modified algorithm, and a detailed description is provided in the Appendix A. 

For each of the four subsets selected from each of the five parent datasets, three EOA models were derived using 7, 10 and 13 descriptors. In addition, we developed five more models, one for each target, using three descriptors selected from an initial pool of seven descriptors. This was done in order to test whether good models could be generated from a more limited set of descriptors which focuses on the description of protein–ligand interactions and key ligand ADME properties. The specific descriptors for the initial pool were selected based on the data in Appendix A (descriptors occurrences) and included PEOE3, HBD, MR1, ALOGP2, ALOGP7, ssCH2_Cnt and aaCH_Cnt (see Appendix A for short explanations of these descriptors). Thus, a total of 5 × 4 × 3 + 5 = 65 EOA models were derived in this work. A typical MC/SA run for the derivation of an EOA model consisted of 1,000,000 MC steps. Simulated annealing was implemented by means of a saw-tooth procedure whereby repeated annealing cycles were performed. In each cycle the RT term was linearly decreased from 1 to 0.01 in 0.01 intervals, running 400 MC steps per interval. The range of values of the RT term led to an acceptance rate of roughly 2–6%.

### 4.3. Molecular Docking 

We chose to compare the EOA algorithm with some of the docking tools commonly used for VS, including AutoDock Vina [10] using the PyRx platform version 0.9 [62], Glide-SP [9] and GOLD [63]. All docking calculations were performed with default parameters. More specifically: For Glide, input conformations for docking were generated using the “Canonicalize input conformation” option, and the energy window for ring sampling was set to 2.5 kcal/mol. The best 400 poses per ligand were kept for energy minimization using the OPLS3e force field and the pose with the lowest, standard precision Glide gscore was kept. For GOLD, the population size for the genetic algorithm was set to 100 and the maximal number of operations per ligand to 100,000. The pose with the lowest CHEMPLP value was kept. For AD Vina, the exhaustiveness argument was set to 8, and the maximum number of binding modes to generate was set to 9. The pose with the highest Vina score was kept. 

In the interest of performing a fair comparison, docking was performed on those compounds included in the test sets used for the evaluation of the EOA models. For each target, all test sets were docked against the two crystal structures (see Table 5). 

### 4.4. Evaluation Metrics

The performances of EOA and of the different docking methods were evaluated using two common metrics, namely, area under the ROC curve (AUC) and enrichment at 1% of the library (EF_1%_). AUC measures the overall performances of the VS procedure, whereas EF_1%_ measures performances at early stages of the screening process. In the case of docking, for each target, the two metrics were calculated for each PDB structure separately. In addition, we implemented a consensus approach whereby each ligand was ranked based on its best docking score across the two structures and the metrics calculated from this consensus ranking.

## Figures and Tables

**Figure 1 ijms-23-00043-f001:**
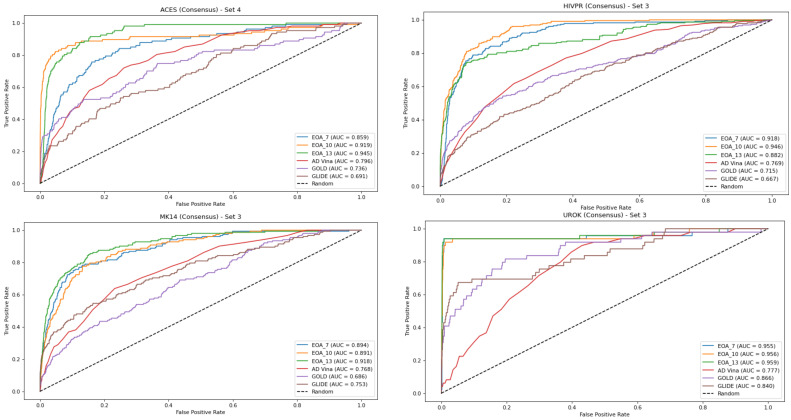
ROC curves for the best-performing docking method based on the consensus approach compared with EOA and the other docking methods (see text for more details). In all cases, the EOA performed better than all docking methods.

**Figure 2 ijms-23-00043-f002:**
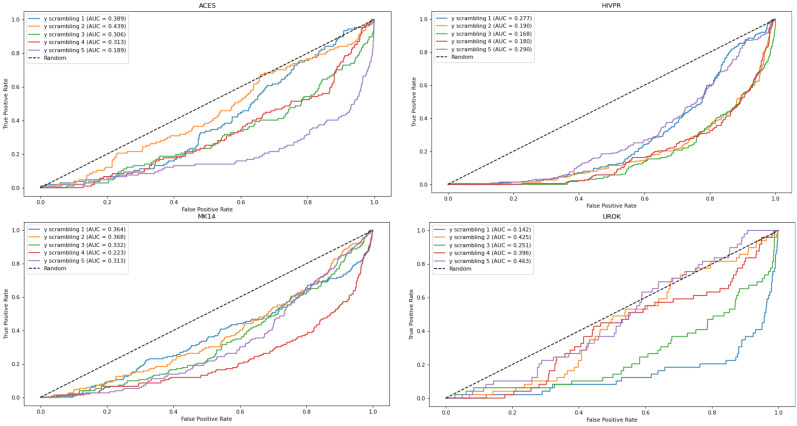
ROC curves obtained from the EOA models derived from the scrambled sets. In all cases AUC values are below 0.5, indicating performances lower than random.

**Figure 3 ijms-23-00043-f003:**
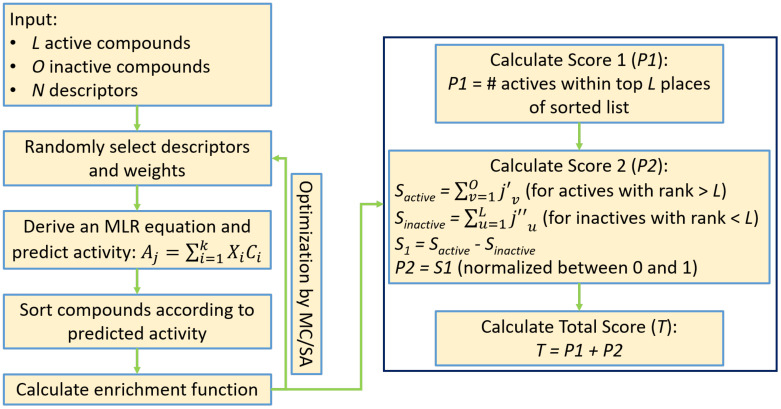
A flowchart of the modified enrichment optimizer algorithm (EOA). See the Appendix A for more details.

**Table 1 ijms-23-00043-t001:** EOA results obtained for all four subsets from all five datasets using 7, 10 and 13-descriptor models. Results are provided in terms on the number and percentage (based on the total number of actives) of active compounds appearing within the first L places of the list ranked according to the EOA equation.

Dataset	# Descriptors	# Actives = *L*			# Actives among *L* top Places		
		Train	Validation	Test	Train (%)	Validation (%)	Test (%)
ACES-1	7	430	106	107	324 (75%)	75 (70%)	15 (14%)
	10				315 (73%)	72 (67%)	28 (26%)
	13				343 (80%)	84 (79%)	49 (46%)
ACES-2	7				312 (73%)	68 (64%)	42 (40%)
	10				326 (76%)	72 (67%)	38 (36%)
	13				357 (83%)	79 (74%)	0 (0%)
ACES-3	7				322 (75%)	78 (73%)	38 (36%)
	10				319 (74%)	74 (69%)	13 (12%)
	13				323 (75%)	69 (64%)	20 (19%)
ACES-4	7				331 (77%)	73 (68%)	47 (44%)
	10				326 (76%)	73 (68%)	56 (53%)
	13				329 (77%)	75 (70%)	5 (5%)
HIVPR-1	7	912	227	227	766 (84%)	187 (82%)	66 (29%)
	10				806 (88%)	204 (90%)	49 (22%)
	13				831 (91%)	202 (89%)	127 (56%)
HIVPR-2	7				742 (81%)	182 (80%)	75 (33%)
	10				830 (91%)	201 (89%)	91 (40%)
	13				801 (88%)	196 (86%)	38 (17%)
HIVPR-3	7				750 (82%)	192 (85%)	66 (29%)
	10				800 (88%)	200 (88%)	73 (32%)
	13				837 (92%)	206 (91%)	124 (55%)
HIVPR-4	7				759 (83%)	182 (80%)	61 (27%)
	10				804 (88%)	197 (87%)	65 (29%)
	13				807 (88%)	198 (87%)	94 (41%)
MK14-1	7	608	151	152	455 (75%)	111 (74%)	41 (27%)
	10				477 (78%)	116 (77%)	40 (26%)
	13				464 (76%)	110 (73%)	37 (24%)
MK14-2	7				469 (77%)	106 (70%)	29 (19%)
	10				471 (77%)	111 (74%)	42 (28%)
	13				491 (81%)	111 (74%)	49 (32%)
MK14-3	7				421 (69%)	114 (75%)	43 (28%)
	10				472 (78%)	125 (83%)	41 (27%)
	13				482 (79%)	122 (81%)	48 (32%)
MK14-4	7				440 (72%)	103 (68%)	27 (18%)
	10				466 (77%)	107 (71%)	34 (22%)
	13				470 (77%)	109 (72%)	39 (26%)
UROK-1	7	200	49	49	192 (96%)	46 (94%)	29 (59%)
	10				193 (97%)	48 (98%)	30 (61%)
	13				191 (96%)	47 (96%)	34 (69%)
UROK-2	7				194 (97%)	47 (96%)	22 (45%)
	10				193 (97%)	47 (96%)	30 (61%)
	13				195 (98%)	46 (94%)	29 (59%)
UROK-3	7				192 (96%)	46 (94%)	27 (55%)
	10				180 (90%)	42 (86%)	25 (51%)
	13				193 (97%)	47 (96%)	34 (69%)
UROK-4	7				194 (97%)	46 (94%)	21 (43%)
	10				195 (98%)	46 (94%)	25 (51%)
	13				194 (97%)	46 (94%)	33 (67%)
TRY1-1	7	504	125	126	445 (88%)	100 (80%)	75 (60%)
	10				460 (91%)	109 (87%)	81 (64%)
	13				438 (87%)	105 (84%)	55 (44%)
TRY1-2	7				449 (89%)	116 (93%)	74 (59%)
	10				465 (92%)	111 (89%)	84 (67%)
	13				456 (90%)	114 (91%)	71 (56%)
TRY1-3	7				463 (92%)	113 (90%)	83 (66%)
	10				465 (92%)	113 (90%)	86 (68%)
	13				461 (91%)	110 (88%)	77 (61%)
TRY1-4	7				455 (90%)	111 (89%)	87 (69%)
	10				453 (90%)	105 (84%)	79 (63%)
	13				464 (92%)	110 (88%)	79 (63%)

**Table 2 ijms-23-00043-t002:** EOA and docking results for all test sets expressed in terms of AUC and EF_1%_ values. The docking results are presented as consensuses of two crystal structures per target. For each dataset, the best result is highlighted. We note that EF_1%_ values are indifferent to the order of the active compounds within the first 1% of the library, and it was therefore not unlikely to obtain identical EF_1%_ values from different methods.

Set	Method	AUC					EF_1%_				
		ACES	HIVPR	MK14	UROK	TRY1	ACES	HIVPR	MK14	UROK	TRY1
1	EOA-7	0.862	0.775	0.905	0.997	0.979	36.449	3.965	40.132	77.551	73.810
	EOA-10	0.886	0.946	0.947	0.997	0.996	26.168	20.705	39.474	81.633	80.159
	EOA-13	0.899	0.977	0.927	0.996	0.986	58.879	59.471	25.658	81.633	58.730
	AD Vina	0.764	0.747	0.737	0.749	0.806	10.280	7.048	10.526	4.082	4.762
	GOLD	0.739	0.726	0.676	0.830	0.861	30.841	13.656	12.500	32.653	15.873
	Glide	0.735	0.678	0.743	0.801	0.847	14.953	11.454	15.789	40.816	35.714
2	EOA-7	0.808	0.813	0.902	0.986	0.958	8.411	14.097	20.395	69.388	75.397
	EOA-10	0.885	0.955	0.914	0.978	0.982	43.925	21.586	41.447	81.633	79.365
	EOA-13	0.921	0.927	0.925	0.987	0.966	9.346	19.383	37.500	73.469	71.429
	AD Vina	0.760	0.754	0.753	0.766	0.795	12.150	3.965	8.553	2.041	4.762
	GOLD	0.719	0.687	0.686	0.785	0.849	28.972	12.335	9.868	28.571	23.810
	Glide	0.693	0.612	0.729	0.816	0.832	14.019	8.370	18.421	36.735	42.857
3	EOA-7	0.896	0.918	0.894	0.955	0.982	42.991	9.692	25.000	79.592	76.984
	EOA-10	0.860	0.946	0.891	0.956	0.980	7.477	34.802	23.684	75.510	79.365
	EOA-13	0.895	0.882	0.918	0.959	0.981	21.495	31.718	27.632	89.796	73.016
	AD Vina	0.762	0.769	0.768	0.777	0.786	10.280	7.048	9.868	6.122	3.968
	GOLD	0.710	0.715	0.686	0.866	0.849	27.103	17.621	9.868	34.694	26.190
	Glide	0.687	0.667	0.753	0.840	0.857	14.953	10.132	23.026	42.857	44.444
4	EOA-7	0.859	0.915	0.892	0.980	0.983	14.019	16.300	21.711	61.224	80.159
	EOA-10	0.919	0.922	0.934	0.986	0.982	58.879	13.656	23.026	67.347	76.190
	EOA-13	0.945	0.978	0.934	0.983	0.983	13.084	51.542	36.184	79.592	79.365
	AD Vina	0.796	0.740	0.766	0.765	0.809	8.411	5.727	5.921	6.122	7.143
	GOLD	0.736	0.722	0.663	0.818	0.854	28.972	13.656	7.895	26.531	25.397
	Glide	0.691	0.646	0.728	0.818	0.860	10.280	12.335	16.447	40.816	46.825

**Table 3 ijms-23-00043-t003:** Test set AUC and EF_1%_ values obtained for 3-descriptors EOA models.

Target	AUC	EF_1%_
ACES	0.825	24.299
HIVPR	0.860	18.943
MK14	0.807	13.158
UROK	0.897	20.408
TRY1	0.932	24.603

**Table 4 ijms-23-00043-t004:** Averaged ± SD Euclidean distances between active and decoy compounds for subsets used to derive the best 10-descriptor models for each target. Distances are based on all principle components obtained from PCA.

Target	Set	# Descriptors	Euclidean Distances (Average ± Standard Deviation)	AUC
ACES	4	10	4.34 ± 1.41	0.885
HIVPR	2	10	4.10 ± 1.23	0.955
MK14	1	10	4.90 ± 1.59	0.947
UROK	1	10	4.59 ± 1.30	0.997
TRY1	1	10	4.72 ± 1.32	0.996

**Table 5 ijms-23-00043-t005:** Descriptions of the five datasets used in this work, including the numbers of active and decoy compounds in training, validation and test sets. The UROK dataset had fewer active/decoy compounds listed in DUD-E in comparison with all other datasets.

Dataset	PDB Codes	# Active	# Decoy	Training		Validation		Test	
				# Active	# Decoy	# Active	# Decoy	# Active	# Decoy
**ACES**	1e66, 1acj	643	24,161	430	3333	106	832	107	19,996
**HIVPR**	1xl2, 2pwc	1366	35,071	912	3333	227	832	227	30,906
**MK14**	2qd9, 3o8t	911	34,896	608	3333	151	832	152	30,731
**UROK**	1sqt, 4fue	298	9262	200	1666	49	416	49	7180
**TRY1**	2ayw, 3rxl	755	24,760	504	3333	125	832	126	20,595

## Data Availability

The data that were used in this work were retrieved from DUD-E database (see Methods section) and are freely available [64]. The docking methods that were used in this work are described in the Materials and Methods section and are available as follows: AutoDock Vina, version 1.1.2, is freely available [65]; GOLD, version 2020.3.0, is available to academia at a reduced price [66]; Glide, via Schrodinger package version 2019-2, is commercially available [67]. Canvas program, via Schrodinger package version 2019-2, which was used for descriptors calculation (see Materials and Methods section), is commercially available as well [67]. The last version of the EOA algorithm, 1.3, was implemented and programmed in Python, version 3.6. The full script is provided by the following link: https://drive.google.com/file/d/1gVKmU2Psnl7WV0pEt1hFQcjsKI1ZNx4r/view?usp=sharing (accessed on 18 December 2021).

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
