# Peer review of "A Comparison between Enrichment Optimization Algorithm (EOA)-Based and Docking-Based Virtual Screening"

_ijms, 2021, doi:10.3390/ijms23010043_

Round 1

Reviewer 1 Report

In this well-written manuscript the authors present an improved version of their Enrichment Optimization Algorithm (EOA) which derives multiple-linear-regression equations for virtual screening. The method is tested on DUD-E datasets for five different targets and compared with the performance of three commonly used docking tools. The EOA-derived functions performed very well and consistently better than the docking-based screens. The latter observation is not too surprising given that the EOA-functions are trained on target-specific data, whereas the docking methods were applied as generic tools with standard settings. This, however, is also clearly stated by the authors, who are well aware of this fact and address this point in their critical and clear discussion of the method and its evaluation. The manuscript and its methods will be of great interest to all readers who want to harness experimental (HT-)screening data for
conducting ligand-based virtual screens on the same target. The approach is very clearly presented. Its application should be straightforward for anyone interested. There are only a few minor issues which the authors should consider uopn revision:

- line 387/388: A few more details should be given about the applied docking parameters (how many runs were carried out, how many poses were generated, how was the best pose selected, which scoring function was applied). This should be explicitly stated, not only because "default parameters" may differ from version to version, but in particular because it is tedious for the reader to get this information from the publications (where it is not always complete and up-to-date) or from the manuals/documenation of the programs.

- Table 1: Why is the number of the actives in the test set of the ACES data sets given as "106-107"? According to Table 2, it should be 106.

- SI, Table S4: The EF1% for set 1 of TYR1 is exactly the same (11.905) for GOLD and Glide. The authors should check whether both values are correct.

- SI Table S4: The low EF1% values for EOA-7 (8.411) and EOA-13 (9.346)
on ACES set 2 are surprising. Why does EOA-10 perform so much better on this set (43.925) and why is even docking better here? Looking at Fig. S3 for set 2, EF1% should be approximately the same for EOA-10 and EOA-13, whereas only EOA-7 shows a much lower early enrichment. This does not agree with the data in the Table.

- SI, Table S5: The EF1% values for AD Vina (7.477) and GOLD (21.495) on ACES set 2 reappear for EOA-10 (7.477) and EOA-13 (21.495) on ACES set 3. Is this true?

 - line 418: I was surprised to read that GOLD is free for academia. This may apply to some countries in the framework of the FAIRE programme, but - to the best of my knowledge - not in general. In fact, on the quoted website
(ref. 57), I found the statement "Academic users receive access to all CSD data and software, at around 95% less than industrial users.", which obviously means that some money has still to be paid ...

Reviewer 2 Report

Major issues:
(1) Of course it is accepted that with increasing number of compouds more
descriptors can be used in a QSAR eq. without running into overfitting.
However, a restriction to three or four descriptors typically covers
the essential features that contribute to binding affinity (hydrogen-bonding,
lipophilicity, steric issues), which is supported by looking at Table S2.
Thus I wonder if the results substantially worsen using only three descriptors
in comparison to 7, 10, and 13 descriptors as in Table 1.

(2) Although the bio-targets are rather unrelated to each other, it would
be of interest how "diverse" their corresponding actives and non-active
ligands are, to find out for which situation the presented approach will
work best.
Since the (non-normalized) descriptor values have been calculated for all
ligands it is straight forward to compute their pair-wise Eucledian distance
in the descriptor space. The largest distance found in a given compound set
thus reflects its diversity. 

Minor issues:
line 43-47: A reference (e.g. review article) is needed hold this statement.
line 50-52: What about similarity searches, such as fingerprints, reduced graphs, etc.?
line 71-76: Worth mentioning in this context is Kubinyi's paradoxon that r^2 of 
most test sets are higher than q^2, see Doweyko J.Comput.-Aided Mol.Des. 22 (2008) 81-89.

A final suggestion for future work:
Genetic Algoritms are typically superior to MC/SA in finding global minima. 
